# Post-Embryonic Lateral Organ Development and Adaxial—Abaxial Polarity Are Regulated by the Combined Effect of *ENHANCER OF SHOOT REGENERATION 1* and *WUSCHEL* in *Arabidopsis* Shoots

**DOI:** 10.3390/ijms221910621

**Published:** 2021-09-30

**Authors:** Yoshihisa Ikeda, Michaela Králová, David Zalabák, Ivona Kubalová, Mitsuhiro Aida

**Affiliations:** 1Centre of the Region Haná for Biotechnological and Agricultural Research, Czech Advanced Technology and Research Institute (CATRIN), Palacký University, 78371 Olomouc, Czech Republic; hradilovamichaela@post.cz; 2Laboratory of Growth Regulators, Institute of Experimental Botany AS CR, Palacky University, 78371 Olomouc, Czech Republic; david.zalabak@upol.cz; 3Leibniz Institute of Plant Genetics and Crop Plant Research (IPK), 06466 Gatersleben, Germany; ivona.kubalova@gmail.com; 4International Research Organization for Advanced Science and Technology (IROAST), Kumamoto University, Kumamoto 860-8555, Japan; m-aida@kumamoto-u.ac.jp

**Keywords:** adaxial–abaxial polarity, *Arabidopsis thaliana*, *ENHANCER OF SHOOT REGENERATION 1*, lateral organ, REVOLTA, shoot apical meristem, SHOOTMERISTEMLESS, WUSCHEL

## Abstract

The development of above-ground lateral organs is initiated at the peripheral zone of the shoot apical meristem (SAM). The coordination of cell fate determination and the maintenance of stem cells are achieved through a complex regulatory network comprised of transcription factors. Two AP2/ERF transcription factor family genes, *ESR1/DRN* and *ESR2/DRNL/SOB/BOL*, regulate cotyledon and flower formation and de novo organogenesis in tissue culture. However, their roles in post-embryonic lateral organ development remain elusive. In this study, we analyzed the genetic interactions among SAM-related genes, *WUS* and *STM*, two *ESR* genes, and one of the HD-ZIP III members, *REV*, whose protein product interacts with ESR1 in planta. We found that *esr1* mutations substantially enhanced the *wus* and *stm* phenotypes, which bear a striking resemblance to those of the *wus rev* and *stm rev* double mutants, respectively. Aberrant adaxial–abaxial polarity is observed in *wus* *esr1* at relatively low penetrance. On the contrary, the *esr2* mutation partially suppressed *stm* phenotypes in the later vegetative phase. Such complex genetic interactions appear to be attributed to the distinct expression pattern of two *ESR* genes because the *ESR1* promoter-driving *ESR2* is capable of rescuing phenotypes caused by the *esr1* mutation. Our results pose the unique genetic relevance of *ESR1* and the SAM-related gene interactions in the development of rosette leaves.

## 1. Introduction

When compared to the majority of animals taking a predetermined body plan, the development of terrestrial plants is more plastic and takes place post-embryonically by producing new organs throughout their lifespan. In the shoot, this unique feature is achieved through the coordination of maintenance of stem cells with continuous lateral organ formation. The stem cell niche of the shoot is maintained by the WUSCHEL (WUS)–CLAVATA (CLV) negative feedback loop [1,2]. Although loss-of-function mutations in *WUS* resulted in the formation of an aberrant flat shoot apical meristem (SAM), the *wus* mutant retains an ability to develop vegetative leaves in a stop-and-go mode from either a defective SAM or ectopic meristem (also called lateral shoot meristem) that emerged from the axils of leaves and cotyledons, and eventually gives rise to the formation of inflorescence meristem [3]. To account for this phenotype, the WUS-independent stem cell specification pathway is suggested [4]. It appears that microRNA regulation participates in the WUS-independent stem cell specification pathway [5]. Partial suppression of the *wus* phenotypes was observed when the heterozygous *men1* activation-tagged allele of miR166a was introduced into the *wus* mutant plant [6]. Similarly, the *jabba-1D* (*jba-1D*) gain-of-function dominant mutant caused by overexpression of miR166g displayed pleiotropic phenotypes, such as SAM enlargement, fasciated stem, and the formation of a radial structure [7]. Concomitant with the phenotypes caused by miRNA166g overexpression, combined triple mutations in the miR165/166 target genes, *PHABULOSA (PHB), PHAVOLUTA (PHV)*, and *CORONA (CNA)*, which encode class III *HOMEODOMAIN-LEUCINE ZIPPER TRANSCRIPTION FACTOR* (*HD-ZIP III*), caused meristem enlargement, and in the *phb phv cna wus* quadruple mutant, rosette leaves were more frequently emerged compared to *wus* [4]. On the other hand, another HD-ZIP III family member, *REVOLTA (REV)*, appears to act distinctly. Out of the five HD-ZIP III family members, only *REV* single mutants revealed conspicuous phenotypes that involve defects in lateral meristem (LM) formation [8]. In addition, the *rev-6* null allele enhanced rosette leaf defects in the corresponding *rev-6 stm-2* and *rev-6 wus-1* double mutant plants [8]. Although the post-embryonic role of *PHB*, *PHV*, and *CNA* is not entirely clear [9,10], their role appears to restrict the stem cell population in the WUS-independent stem cell specification pathway whereas *REV* was antagonized by these three *HD-ZIP III* genes [4]. 

The *ENHANCER OF SHOOT REGENERATION 1* (*ESR1*) in AP2/ERF (ETHYLENE RESPONSE FACTOR) transcription family [11], also known as DORNRÖSCHEN (DRN) [12], and its closest homolog, ESR2 [13], also known as DORNRÖSCHEN-LIKE (DRNL) [12], SOB2 (SUPPRESSOR OF PHYTOCHROME B) [14], and BOLITA (BOL) [15] in *Arabidopsis thaliana* and LEAFLESS (LFS) [16] in tomato, were involved in lateral organ emergence [13,17], gynoecium development [18], and stamen enlargement [19]. The organogenesis in *esr1-1 esr2-2* double mutant root explants was largely compromised in a tissue culture system [20] and the analyses of intact double mutant seedling suggested that the disturbed auxin transport is likely responsible for the pleiotropic phenotypes [16,17,20]. We and others identified *CUC1* and *CUC2* as downstream target genes directly regulated by ESR1 [21,22] and ESR2 [13]. Both ESR1 and ESR2 proteins have been documented to physically interact with five members of the HD-ZIP III family (REV, PHB, PHV, CNA, and HB8), both in vitro and in vivo [17,23]. However, the physiological relevance of their interactions remains obscure. Analogous to the genetic interaction between *WUS* and *HD-ZIP III* members, we hypothesize that the two *ESR* genes might have a genetic interaction with *WUS* and *STM*.

Since the roles of the two *ESR* genes in cotyledon and flower development have been documented previously, we exerted our efforts on elucidating the regulatory mechanism of rosette leaf development in the vegetative phase, particularly by scrutinizing the involvement of *ESR* genes in the initiation of rosette leaves from the ectopic/lateral meristem in the SAM-deficient mutant backgrounds (*wus* and *stm*). Our present mutant analyses revealed that, although the *esr1* single mutant did not show phenotypes observed in the *rev* mutant, *esr1* phenocopied *rev* both in the *wus* and *stm* backgrounds to a great extent, whereas *esr2* did not. Rather, in the later vegetative phase, *esr2* partially rescued the retarded rosette leaf development observed in *stm*. Such contradictory observations are reconciled by the fact that the expression of the *ESR2* gene under the control of the regulatory sequences of *ESR1* in *wus esr1* rendered phenotypes indistinguishable from those in the *wus* single mutant, suggesting that the two *ESR* genes have redundant functions but their distinct expression patterns define their physiological relevance in the development of rosette leaves and the establishment of adaxial–abaxial polarity.

## 2. Results

### 2.1. esr1 Mutations Enhanced Defects in Rosette Leaf Development and Adaxial—Abaxial Polarity in wus Background

By 8 days after germination (d.a.g.), successive lateral organ formation leading to the development of at least four recognizable rosette leaves was observed in the wild type (Figure 1A). In terms of continuous emergence of rosette leaves, *esr1-1* was indistinguishable from the wild type (Figure 1B), although, as reported previously, cotyledon phenotypes were observed at low penetrance [13,17,20]. In this study, we identified and characterized a novel *esr1* allele, Gabi Kat 369_A3, where T-DNA insertion is located at 7 bp upstream from the stop codon (Appendix AA). The allele is termed *esr1-2* hereafter (Figure 1C). Endogenous full-length *ESR1* transcripts containing 3′-UTR were absent; however, *ESR1* transcripts lacking 3’-UTR accumulated in *esr1-2* (Appendix AB). Although it is not clear how the truncated *ESR1* transcripts are efficiently translated, *esr1-2* is likely to be a weaker allele than *esr1-1* because the penetrance of the cotyledon phenotypes was lower than that of *esr1-1* and no gain-of-function phenotypes caused by *ESR1* overexpression were observed in *esr1-2* under our growth conditions (data not shown). The development of rosette leaves in *esr2-2* was indistinguishable from that of the wild type (Figure 1D), although the cotyledon phenotypes appeared at low penetrance [13,17]. Phenotypes observed in *esr1-1 esr2-2* were pleiotropic, ranging from the formation of a single cotyledon with delayed emergence of rosette leaves (Figure 1E) to the lack of a hypocotyl with a shorter root, as reported for the *monopteros* (*mp*) mutant (Figure 1F). We confirmed retarded rosette leaf emergence in the two independent *wus* alleles, *wus-1* (Figure 1G) and *wus-101* (Figure 1H). The original *wus-1* in L*er* accession was introgressed into Col-0 (see Materials and Methods). The *WUS* transcript was undetectable in *wus-101* [24]. To gain insight into the physiological relevance of the *ESR* genes in the WUS-independent post-embryonic lateral organ development, *esr1-1* was introduced into the two independent *wus* alleles. Consequently, *wus-1 esr1-1* seedlings exhibited a variety of phenotypes; substantially delayed emergence of rosette leaves (Figure 1I), the formation of a radial structure (Figure 1J–K), and moderate delay in leaf emergence (Figure 1L). Under our growth conditions, no radial structure was found in the *wus* single mutant alleles. We could confirm all the above-mentioned phenotypes in *wus-101 esr1-1* seedlings (Figure 1M–P). Hence, the *wus-101* allele was used for the genetic crosses. Even on 14 d.a.g. approximately 55% of the *wus-101 esr1-1* seedlings did not develop recognizable rosette leaves (Figure 1N and Table 1). We observed weaker enhancement of lateral organ phenotypes in *wus-101 esr1-2*, resulting in the formation of a radial structure at a lower frequency than *wus-101 esr1-1* (Figure 1Q) and intermediate rate of rosette leaf emergence between *wus-101* and *wus-101 esr1-1* (Figure 1R and Table 1). By 10 d.a.g., 97.9% of the *wus-101 esr1-2* seedlings were capable of developing at least one rosette leaf or radial structure (Table 1). The contribution of *ESR2* in the WUS-independent rosette leaf development was incomparable with that of *ESR1* because the *esr2-2* mutation subtly enhanced the *wus* phenotype up to 10 d.a.g. (Figure 1S-T and Table 1). No radial structure was observed in *wus-101 esr2-2* under our growth conditions (Table 1). In the *wus-101 esr1-1 esr2-2* triple mutant, in addition to phenotypes found in *wus-101 esr1-1*, around 21% of the seedlings did not produce fully developed and differentiated cotyledon and rosette leaves (Figure 1U–V), and immaturely died later. By 30 d.a.g., the soil-grown *wus-101 esr1-1* adult plant only developed a pair of fully expanded rosette leaves, whereas the *wus-101 esr1-2* plant developed rosette leaves more frequently compared to the *wus-101 esr1-1* plant (Figure 1W). Approximately 32% of *wus-101 esr1-2* formed at least one lotus-like rosette leaf (Figure 1W inset). In the case of *wus-101 esr1-1 esr2-2* triple mutants, 34.7% of them failed to develop rosette leaves. Instead, a mass of undifferentiated and disorganized cells accumulated in the shoot apex (Figure 1X) or in the ectopic meristem that emerged beneath the original SAM (Figure 1Y). 

### 2.2. wus-101 esr1-1 Phenocopied wus-101 rev-5

The consistently observed defects in rosette leaf development of seedlings with the different *wus* and *esr1* allele combinations (Figure 1I–R) bear a striking similarity to those of the *wus-1 rev-6* double mutant in L*er* [8]. Besides, the REV protein reportedly interacts with ESR1 [23] and ESR2 [17], although the interaction between REV and ESR2 remains a matter of debate [25]. To study the genetic interaction between *REV* and two *ESR* genes, *rev-5* in Col-0 accession was used for this purpose [8]. In the case of successive emergence of rosette leaves, the *rev-5* seedling was indistinguishable from the wild type (Figure 2A and Figure 3C). Consistent with the previous results [8], we were able to confirm substantial enhancement of the *wus* phenotypes by the *rev-5* mutation in the corresponding *wus-101 rev-5* (Figure 2C). *wus-101 rev-5* double mutant seedlings formed a radial structure more frequently than *wus-101 esr1-1* (Figure 2D,I), whereas mutations of its close homologs, *phb* and *phv*, did not (Figure 2E,F). A novel T-DNA insertion allele of *PHB*, SALK_008924C, in which a single T-DNA is inserted into exon 7 (2045 bp downstream from the ATG codon) (Appendix A), was employed for crossing with *wus-101*. This novel *phb* allele is termed *phb-101* hereafter. When *rev-5* and *esr1-1* mutations were combined in *wus-101*, the resulting *wus-101 rev-5 esr1-1* triple mutant phenotypes appeared to be enhanced in an additive manner (in comparison to the respective double mutants). Unlike the *esr1-1* mutation, *esr2-2* in *wus-101 rev-5* affected in a developmental stage-dependent manner. Until 10 d.a.g., *wus-101 rev-5 esr2-2* seedlings failed to develop rosette leaves more frequently than *wus-101 rev-5* (Figure 2H), whereas such an enhanced phenotype was mitigated by 17 d.a.g. (Figure 2I).

### 2.3. esr1-1 and esr2-2 Antagonistically Regulate Rosette Leaf Development in bum1-3 in the Later Vegetative Phase

Previously, the *rev-6* mutation has been shown to enhance *stm-2* phenotypes both in intact plants and in tissue culture [8]. We sought for the role of two *ESR* genes in the successive development of rosette leaves in the *stm* background. A weak allele of *SHOOTMERISTEMLESS/BUMBERSHOOT1 (BUM1)* in the Col-0 accession, *bum1-3*, was used [26]. Similar to *wus* seedlings (Figure 1G,H), *bum1-3* exhibited a discontinuous rosette leaf emergence (compare Figure 3A,B). Similar to the wild type, both *rev-5* and *rev-5 esr1-1* seedlings were capable of developing true leaves continuously (Figure 3C,D). As shown in the previous study [8], we observed consistent phenotypes in rosette leaves of *bum1-3 rev-5* double mutant seedlings: pronounced delay of rosette leaf emergence (Figure 3E), formation of a radial structure (Figure 3F), aberrant cotyledon in size and shape with delayed emergence of rosette leaves (Figure 3G), and the formation of a pin structure (Figure 3H). Similarly, *bum1-3 esr1-1* had delayed emergence of rosette leaves (Figure 3I,J). At 3.34% frequency, the cotyledon was completely fused (Figure 3K). Note that on 17 d.a.g., a rosette leaf developed from the shoot apex, suggesting that the SAM still retained its activity to develop a rosette leaf although the emergence was substantially delayed. Under our growth conditions, we did not find *bum1-3 esr1-1* seedlings forming a radial structure. Or the penetrance is too low to discover in *bum1-3 esr1-1.* The emergence of rosette leaves in *bum1-3 esr2-2* seedlings was affected in a developmental stage-dependent manner. In the case of continuous rosette leaf emergence, the *esr2-2* mutation enhanced the *bum1-3* phenotype until 10 d.a.g. (Figure 3L). Later on, *bum1-3 esr2-2* seedlings developed true leaves more effectively than the *bum1-3* single mutant seedlings (*p* < 0.001, *n* > 50; Figure 3Q), showing that the *esr2-2* mutation suppressed the *bum1-3* phenotype in the later vegetative phase. On the other hand, introducing either *rev-5* or *esr2-2* into the *bum1-3 esr1-*1 background weakly enhanced aberrant lateral organ phenotypes (Figure 3M–P). The number of developed rosette leaves of *bum1-3 rev-5* was indistinguishable from that of *bum1-3 esr1-1* (*p* > 0.1, *n* > 60), suggesting that, consistent with results obtained from the *wus-101* background, *ESR1* plays a role in successive rosette leaf emergence in the same manner as *REV* does, presumably by forming a protein complex to modulate gene expression in the STM/BUM-independent pathway. *bum1-3 rev-5 esr1-1* seedlings showed a wide range of phenotypes; from relatively milder enhancement (Figure 3M) to aberrant development (Figure 3N). The number of developed rosette leaves on 21 d.a.g. in *bum1-3 rev-5* or *bum1-3 esr1-1* double mutant seedlings was 2.92 ± 1.91 or 3.15 ± 1.34, respectively, whereas in the *bum1-3 rev-5 esr1-1* triple mutant it was 2.51 ± 1.33 leaves. Since both *bum1-3 rev-5* and *bum1-3 esr1-1* double mutants exhibited a severe phenotype, the triple mutant did not statistically differ from the respective double mutants (*p* > 0.1 in both cases, *n* > 50; Figure 3Q).

### 2.4. Distinct Expression Pattern of ESR Genes Defines their Unique Roles

In both the *wus* (Figure 4A) and *bum* mutant backgrounds, *rev-5* and *esr1-1* similarly enhanced defects in rosette leaf development (Figure 4B), whereas *esr2-2* had an opposite effect in the later vegetative phase (beyond 10 d.a.g.). Nevertheless, *ESR1* and *ESR2* are the closest homologs and cause similar phenotypes when overexpressed: cytokinin-independent shoot regeneration in the tissue culture and the accumulation of undifferentiated cells [11,13]. They also share the same downstream target genes [13,21]. These findings suggest that, in terms of regulating downstream gene expression, they are comparable with each other. To tackle this discrepancy, we hypothesize that the distinct expression pattern of the two *ESR* genes is responsible for such contradicting results. To corroborate the spatial and temporal *ESR1* expression, we have identified a *GUS* enhancer trap line, termed *ESR1en:GUS*, whereby the reporter is driven under the influence of an endogenous *ESR1* locus. In this line, a single copy of pD911 T-DNA that contains a -60 Cauliflower mosaic virus minimal promoter fused to the *uidA* (*GUS)* reporter gene [27] is inserted at 73 bp upstream from the ATG codon that corresponds to the putative transcription start site. The right border is oriented toward the *ESR1* promoter (Figure 4C). Using this line, we confirmed the consistent expression pattern of *ESR1* in the upper layers in the CZ and PZ of the SAM (Figure 4D), as reported previously [17]. On the other hand, the expression of *ESR2* is predominantly enriched in the founder cells of leaf primordia in the early vegetative phase [13,28]. The expression pattern of ESR1, ESR2, WUS, STM, and REV in the vegetative shoot has been reported previously [13,29,30,31,32] and their protein distribution is schematically represented (Appendix A). The binary vector harboring the *ESR1* promoter-driving *ESR2* coding sequence containing the 2.86 kb *ESR1* downstream region (with *ESR1* 3′-UTR included) was introduced into the *wus-101* +/− *esr1-1* −/− genotype. In the T3 generation, four independent lines homozygous for the transgene, termed *pESR1:ESR2_ESR1* 3′-UTR, in the *wus-101 esr1-1* double mutant background, developed rosette leaves in the same manner as the *wus-101* single mutant does, demonstrating that loss of *ESR1* functions can be replenished by *ESR2* driven by the *ESR1* regulatory sequence and that the two *ESR* genes have redundant functions (Figure 4F).

## 3. Discussion

In this study, we employed the *esr1-1/drn-2* and *esr2-2* alleles because, unlike *drn-1 drnl-2* (null allele combination), *esr1-1 esr2-2* double mutant plants still produce a small number of viable seeds, which enabled us to examine genetic interactions with *wus* or *bum* and to examine the corresponding triple mutant rosette leaf phenotypes. In the case of lateral organ formation phenotypes on 8 d.a.g., *wus-101* appears to exhibit a stronger phenotype than that of *wus-1*, indicating that *wus-101* is a null allele (Table 1). It is intriguing that the lateral organ phenotypes observed in the *wus esr1* double mutant combinations bear a striking resemblance to those in *wus-1 rev-6* [8], albeit the fact that single *esr1* mutant alleles examined so far do not exhibit phenotypes found in *rev* single mutants. The same holds true in the case of the *bum1-3* mutant background that the number of developed rosette leaves of *bum1-3 rev-5* is indistinguishable from that of *bum1-3 esr1-1* (Figure 3Q). These results support the notion that the ESR1 and REV proteins physically interact with each other to control axillary meristem formation [23]. In fact, the overlapping expression of *ESR1* and *REV* in leaf primordia was shown [23]. It is noteworthy that, unlike *REV* expression confined within the adaxial region of developing leaves [7], the expression pattern of *ESR1* in young leaf primordia is broader [23]. Although it is not clear yet how the *esr1* mutation operates to establish adaxial–abaxial polarity in the *wus* mutant background, we repeatedly observed the adaxial–abaxial polarity defects in rosette leaves of various *esr1 wus* double mutant backgrounds (Figure 1J,K,O and Table 1) at a lower frequency than in *wus-101 rev-5* (Figure 2I). Besides, lotus-like rosette leaves are more frequently found in *wus-101 esr1-2*, a hypomorphic *esr1* allele we introduced in this study, than in *wu1-101 esr1-1* (Figure 1W), implying that lotus-like rosette leaf is formed due to the milder adaxial–abaxial polarity defects. The same structure was reported previously in 12.5% of *as2-101* single and 23.5% of *rev-6 as2-101* double mutant plants [33]. In the same work, the authors also found a needle-like leaf, which resembles what we call a radial structure (Figure 2D), among *as2-101 rev-6*, *as2-101 phb-6*, and *as2-101 phv-5* double mutants [33]. It is noteworthy to mention that defects in the adaxial–abaxial polarity observed in *wus-1 rev-6* and *as2-101 rev-6* are L*er* accession and that, in the case of the *as1* and *as2* backgrounds, *erecta* (*er*) mutation facilitates leaf polarity defects [34,35]. It is interesting to examine the genetic interaction between *ESR1* and *ER* in the future.

The fact that radial structure formation was more frequently observed in *wus-101 esr1-1* than in *wus-101 esr1-2,* a weak *esr1* allele, suggests that, in concert with REV by physical protein–protein interaction, WUS-independent rosette leaf emergence is modulated in an *ESR1* dosage-dependent manner (Figure 5). Recently, Xu and colleagues found the remarkably enriched expression of *WUS* and *ESR1/DRN* during the regeneration period in mesophyll protoplast regeneration culture and that both of which are required for somatic cell regeneration [36]. The interplay between *WUS* and *ESR1* is implicated and our present genetic results are in agreement with them. It appears that *ESR1* genetically interacts with other factors because the previous work showed the aberrant development of rosette leaves in *pcn (popcorn) drn-1* double mutant [37].

Although *ESR1* and *ESR2* have redundant functions and exhibit similar cotyledon phenotypes [13,17,20], *ESR1,* in concert with *WUS* and *STM,* appears to play more important roles in lateral organ emergence and the establishment of adaxial–abaxial polarity.

Our result that the *wus-101 esr1-1* double mutant transformed with the construct harboring the *ESR1* promoter-driving *ESR2* is indistinguishable from *wus-101* (Figure 4F) corroborates that the two ESR proteins have redundant functions and are fungible. Reciprocally, the compromised shoot regeneration phenotype of the *esr2-2* root explants in the tissue culture system was rescued by the *ESR2* promoter-driving *ESR1* [20]. These results suggest that the two *ESR* genes respond differently to internal and external cues. Yet, it is intricate to interpret the fact that the *esr2* mutation partially rescued the inconsistent rosette leaf emergence in *bum1-*3 (Figure 3Q) and in *wus-101 rev-5* (compare *wus-101 rev-5* with *wus-101 rev-5 esr2-2* on Day 17 in Figure 2I) in the later vegetative phase. Unexpectedly, *wus-101 esr1-1 esr2-2* triple mutants accumulated numerous undifferentiated cells at the shoot apex (Figure 1X), and, as a consequence, no rosette leaves were differentiated. Monitoring the SAM marker gene expression in the triple mutant shoot apex is anticipated in the future study.

## 4. Materials and Methods

### 4.1. Plant Material and Growth Condition

*Arabidopsis thaliana* accession Columbia-0 (Col-0) was used as the wild type. The seeds described below were obtained from the European Nottingham Arabidopsis Stock Centre (NASC)): *esr1-1/drn-2* (N121728) [17,20] *wus-101* (N483520) [24], *wus-1* (N15) [38], *phv* (N862830) [17], *phb-101* (N654985), *esr1-2* (N321463), and *bum1-3* (N3781). Homozygous seeds of *esr2-2* were kindly obtained from Hiroharu Banno [20]. *rev-5* (Col-0 accession) was originally isolated in Luca Comai’s lab and homozygous seeds were kindly obtained through Ida Ruberti [39]. Prior to making higher-order mutants, all mutants employed in this work were backcrossed at least four times and the original *wus-1* (L*er* accession) was introgressed into Col-0 through repetitive crossing with Col-0 six times. Mutations were genotyped by PCR by a conventional method. For genotyping *esr2-2, wus-1*, *bum1-3*, and *rev-5,* dCAPS markers were developed and the respective PCR products were digested with *Eco*RV, *Nco*I, *Cla*I, or *Sna*BI, respectively. Primers used for the genotyping are listed in Appendix A. The GUS enhancer reporter line, *ESR1en:GUS*, was obtained by PCR-based screening [27] and additional T-DNA insertions present in the original *ESR1en:GUS* were segregated out by repetitive backcrossing with Col-0 five times. A single pD991 T-DNA insertion in the *ESR1* locus was confirmed by kanamycin segregation analysis and Southern blotting. Primers used for the screening and confirming the insertion position of T-DNA are listed in Appendix A. Seeds were surface-sterilized, sown on MS plant agar medium, and grown at 21 °C in a photoperiod of 16/8 (light/dark) condition at the indicated days. Clearing of seedlings and photographing differential interference contrast (DIC) were carried out as described previously [40]. Seedlings were photographed by a Stereoscopic Zoom Microscope SMZ1000 (Nikon, Tokyo, Japan) operated with NIS Elements software at the indicated time points.

### 4.2. Construction of Transgene and Transformation

The binary vector, pHLG60, a modified version of pSK34 [13], contains a hygromycin resistant cassette for plant transformation. The *ESR1* promoter, its downstream region, and *ESR2* coding sequence were PCR-amplified from Col-0 genomic DNA as a template by using Phusion^®^ High-Fidelity DNA Polymerase (NEB, Massachusetts, USA). The primers used are listed in Appendix A. The resultant PCR products were digested with *Asc*I and *Bam*HI (*ESR1* promoter), *Bam*HI and *Spe*I (*ESR2* ORF), and *Spe*I and *Not*I (*ESR1* downstream region) and sequentially cloned into pHLG60 at the corresponding restriction endonuclease recognition sites. The resulting construct was introduced into *Agrobacterium tumefaciens* strain GV3101, which was used to transform the *wus-101* +/- *esr1-1* -/- genetic background by the floral dip method. Harvested seeds were plated on MS agar medium containing 18 mg/L hygromycin B for the transgene and 2 mg/L sulfadiazine sodium salt for *wus-101* selections. Four independent T3 lines containing a single insertion for the transgene in the *wus-101* +/- *esr1-1* -/- background were chosen for the analysis.

### 4.3. Semi-Quantitative RT-PCR Analysis

Conditions for RNA extraction, first-strand cDNA synthesis, PCR, and agarose gel electrophoresis were previously described [13]. Primers for detecting *ESR1* transcripts in *esr1-2* are listed in Appendix A and those for *TUBULIN3* were described [13]. Cycles used for detection of *ESR1* or *TUB3* are 31 or 18 cycles, respectively.

## Figures and Tables

**Figure 1 ijms-22-10621-f001:**
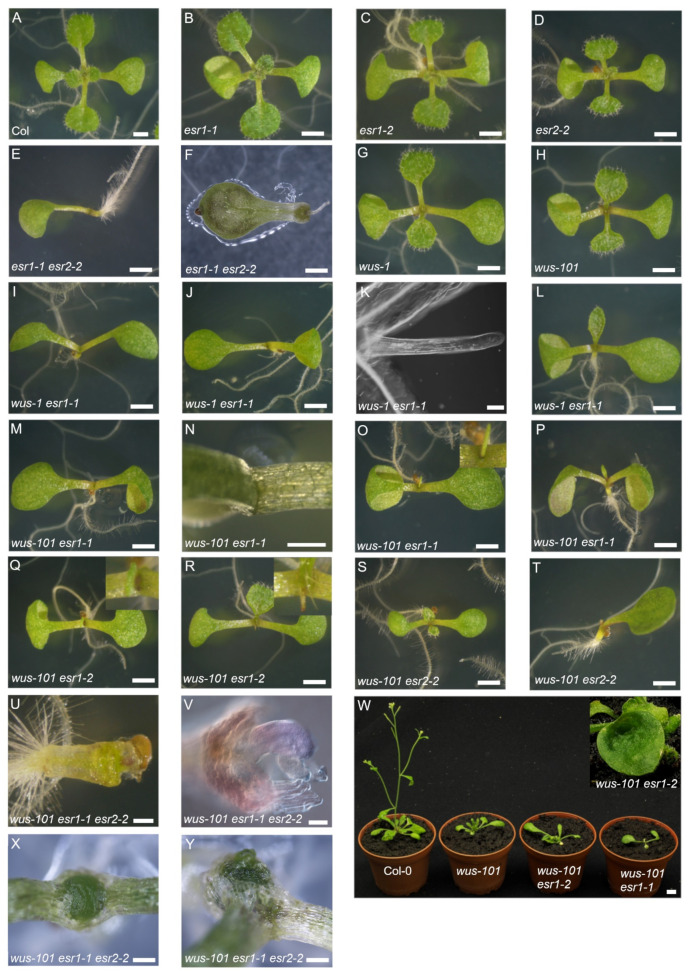
*ESR1* controls rosette leaf development in the WUS-independent pathway in a dosage-dependent manner. (**A**) Eight-day-old Col-0; (**B**) *esr1-1*; (**C**) *esr1-2*; (**D**) *esr2-2*; (**E,F**) *esr1-1 esr2-2*; (**G**) *wus-1*; (**H**) *wus-101*; (**I**–**L**) *wus-1 esr1-1*; (**K**) Normarski image of the plant (**J**); (**M**–**P**) *wus-101 esr1-1*; (**N**) fourteen-day-old *wus-101 esr1-1,* note the radial structure (inset) in (**O**); (**Q**,**R**) eight-day-old *wus-101 esr1-2,* note the single radial structure (inset); (**S,T**) *wus-101 esr2-2*; (**U**) *wus-101 esr1-1 esr2-2*; (**V**) Nomarski image of the shoot apex of seedling (**U**); (**W**) thirty-day-old plants of Col-0, *wus-101, wus-101 esr1-2,* and *wus-101 esr1-1* (from left to right), note the lotus-like leaf of *wus-101 esr1-2* (inset); (**X**,**Y**) thirty-day-old *wus-101 esr1-1 esr2-2* shoot apex. Scale bars = 1 mm (A–E, G–J, L–T,W), 0.5 mm (**F,U,X,Y**), and 0.2 mm (**K,V**).

**Figure 2 ijms-22-10621-f002:**
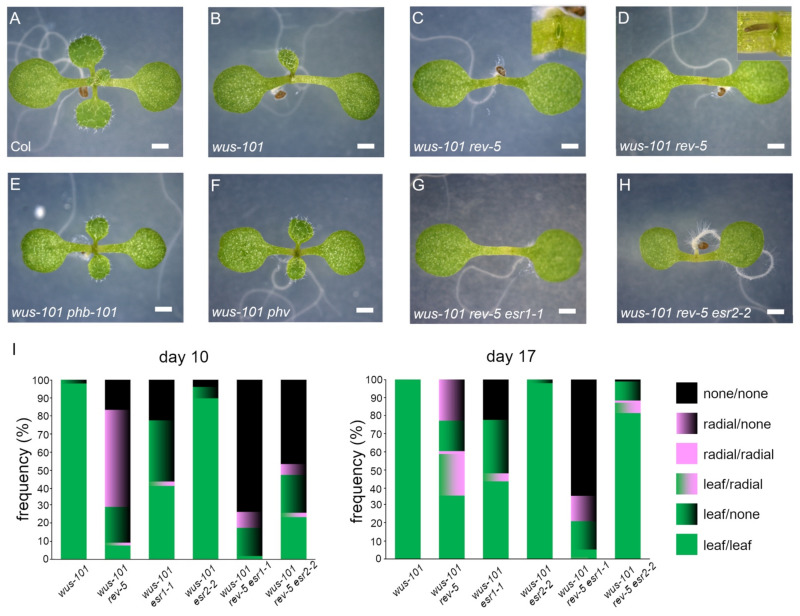
*rev-5* but not *phb-101* or *phv* enhances *wus* phenotype. (**A**) Five-day-old Col-0; (**B**) *wus-101*; (**C,D**) *wus-101 rev-5*, note the radial structure (inset) in (**D**); (**E**) *wus-101 phb-101*; (**F**) *wus-101 phv*; (**G**) *wus-101 rev-5 esr1-1*; (**H**) *wus-101 rev-5 esr2-2*; (**I**) frequency of phenotypes in the respective mutant background on 10 d.a.g (left) and 17 d.a.g. (right). At least 50 individual plants per genotype were examined with biological triplicates. The first pair of emerged rosette leaves are categorized into a developed leaf (green) or radial structure (pink). Black indicates no development of lateral organs at the indicated time points.

**Figure 3 ijms-22-10621-f003:**
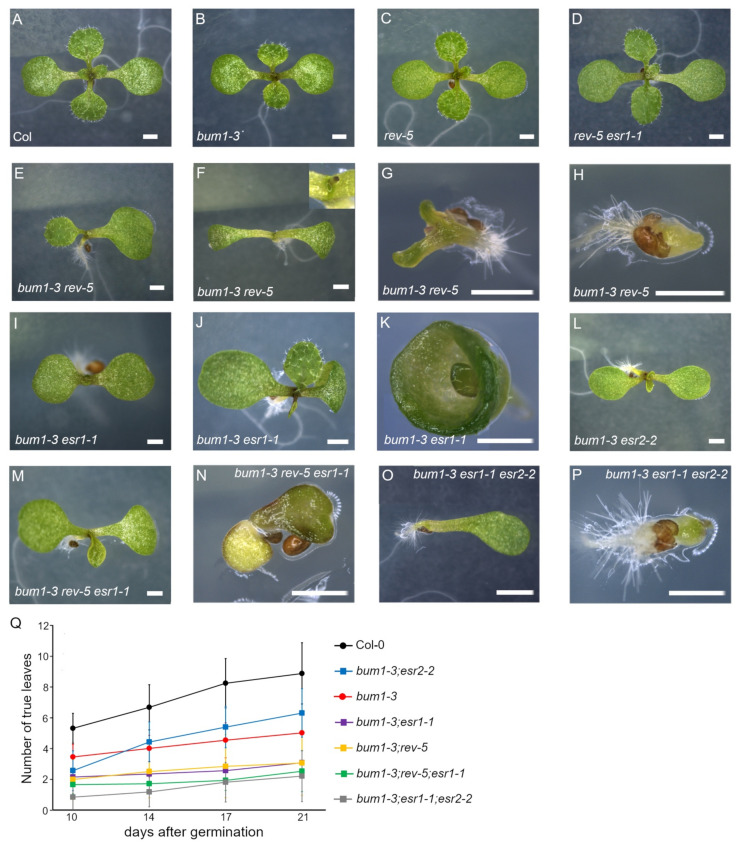
*esr1* and *rev* additively enhance the *bum* phenotype. (**A**) Eight-day-old Col-0; (**B**) *bum1-3*; (**C**) *rev-5*; (**D**) *rev-5 esr1-1*; (**E**–**H**) *bum1-3 rev-5*, note the single cotyledon in (**E**) and radial structure (inset) in (**F**); (**I**–**K**) *bum1-3 esr1-1*; (**L**) *bum1-3 esr2-2*; (**M,N**) *bum1-3 rev-5 esr1-1*; (**O,P**) *bum1-3 esr1-1 esr2-2*; (**Q**) number of developed rosette leaves in the respective mutant backgrounds at 10, 14, 17, and 21 d. a. g. Data shown are the mean ± SD of biological triplicates (*n* > 50 per genotype). Scale bars = 1 mm.

**Figure 4 ijms-22-10621-f004:**
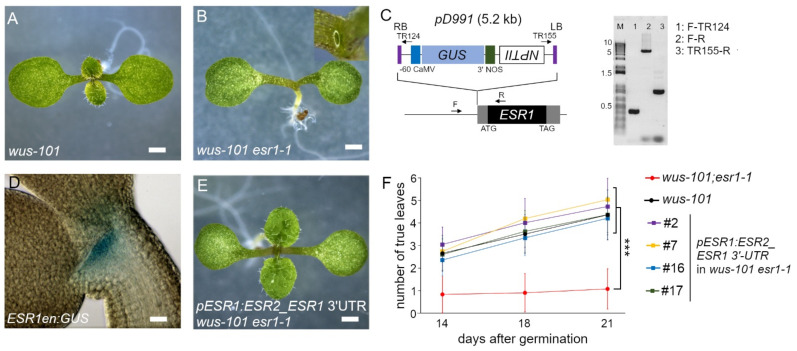
*ESR1* promoter-driving *ESR2* complements *esr1* phenotypes in *wus-101 esr1-1*. (**A**) Eight-day-old Col-0. (**B**) *wus-101 esr1-1*. (**C**) Schematic representation of the *ESR1* enhancer trap *GUS* reporter line (left) and the determination of T-DNA insertion position and orientation. A single copy pD99 T-DNA is inserted at 73 bp upstream from the ATG codon facing the right border toward the *ESR1* promoter. Note the correct orientation of the *uidA* (*GUS*) transgene. Arrows indicate the primers used for screening and verifying the T-DNA insertion. M, marker. Black, gray, blue, right blue, green, purple, and open rectangle indicate *ESR1* coding, *ESR1* UTR, -60 Cauliflower mosaic virus minimum promoter, *uidA* (*β-GLUCURONIDASE*) coding, *NOS* terminator, right or left border, and *NPTII* (*NEOMYCIN PHOSPHOTRANSFERASE II*), respectively. (**D**) Histological GUS staining of *ESR1* enhancer trap line on 2 d.a.g. (**E**) Eight-day-old *wus-101 esr1-1* double mutant seedlings harboring the homozygous *pESR1:ESR2_ESR1* 3′-UTR transgene. (**F**) Number of developed rosette leaves. Four independent transgenic lines harboring a transgene (homozygous single insertion) in the *wus-101 esr1-1* background and their parent, *wus-101 esr1-1*, were compared at the indicated time points. *wus-*101 was included as a positive control. Data shown are the mean ± SD of biological triplicates (*n* > 50) (*p* < 0.001; Student *t*-tests).

**Figure 5 ijms-22-10621-f005:**
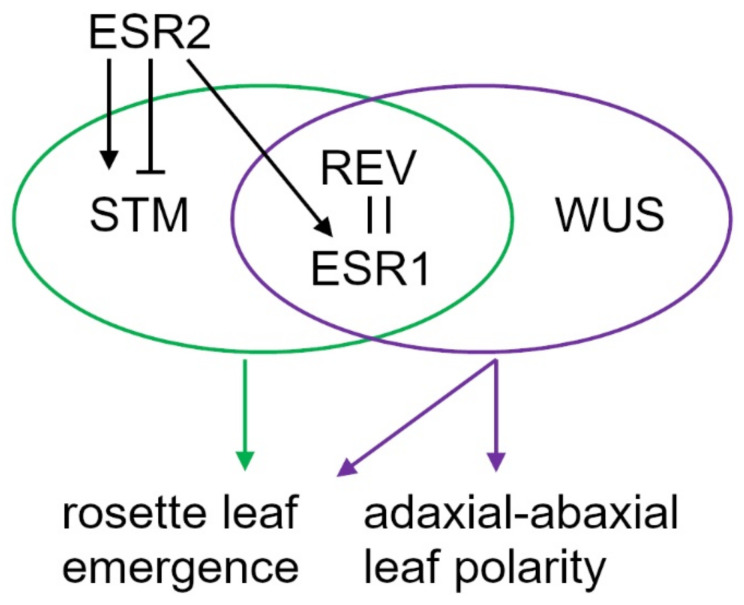
Schematic representation of genetic interactions. The REV and ESR1 protein interaction is confirmed. Together with STM or WUS, both ESR1 and REV participate in the rosette leaf emergence and the establishment of adaxial–abaxial polarity. ESR2 and ESR1 are functionally interchangeable whereas the effect of ESR2 on rosette leaf emergence in the *stm/bum* mutant is vegetative phase-dependent.

**Table 1 ijms-22-10621-t001:** Frequency of variable shoot phenotypes (%) on day 10 after germination ^a^.

	*wus ^b^*	Radial Structure ^d^	Unrecognizable True Leaves	*mp*-Like	pin-Like Shoot
*wus-1* (*n* = 160)	100	0	0	0	0
*wus-1;esr1-1* (*n* = 54)	68.5 ^c^	14.8	14.8 ^e^	1.9	0
*wus-101* (*n* = 196)	100	0	0	0	0
*wus-101;esr1-1* (*n* = 88)	35.2 ^c^	9.1	55.7 ^e^	0	0
*wus-101;esr1-2* (*n* = 38)	78.9 ^c^	21.1	0	0	0
*wus-101;esr2-2* (*n* = 55)	97.9	0	2.1	0	0
*esr1-1;esr2-2*^e^ (*n* = 47)	0	0	0	26.1	0
*wus-101;esr1-1;esr2-2* (*n* = 72)	9.7 ^c^	5.6	34.7 ^f^	38.9	11.1

a: Cotyledon phenotypes are not counted; b: *wus* phenotype denotes reduced number of developed true leaves; c: *wus* phenotypes are moderately enhanced; d: Includes seedling developing at least one radial structure; e: Only two fully expanded leaves are developed by 40 d.a.g; f: No visible true leaves developed by 40 d.a.g.

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
