# Peer review of "Post-Embryonic Lateral Organ Development and Adaxial—Abaxial Polarity Are Regulated by the Combined Effect of ENHANCER OF SHOOT REGENERATION 1 and WUSCHEL in Arabidopsis Shoots"

_ijms, 2021, doi:10.3390/ijms221910621_

Round 1

Reviewer 1 Report

The authors present an extensive genetic analysis aimed at understanding the WUSCHEL and STM independent regulation of shoot meristem development. In this context, they analyze the possible roles of ESR family and HD-ZIP family transcription factors. Though studies on WUS and STM indicate their direct roles in SAM development. However, the roles of ESR and HDZIP genes and their interactions with WUS or STM have remained less explained mainly due to genetic redundancy. In this context, authors carry out genetic interactions between wus, stm, the two esr genes and the rev which belongs to HD-Zip family focusing on the post embryonic development. Their work reveals that one of the ESR gene (ESR1) enhanced wus and stm mutant phenotypes which resembles esr1;rev double mutant phenotypes. However, ESR2 was able to partially rescue stm mutant phenotype. Authors through rescue experiments show that both ESR genes have analogous function but differences in their expression patterns could have led to differences observed in genetic interactions with STM. My major concern is that authors do not provide expression pattern of ESR2. Has it been published elsewhere? If so authors must include a schematic and explain how such expression pattern contributed to the rescue of stm mutant phenotype? In general, though genetic analysis is thorough, I struggle to connect this analysis with the possible molecular circuitry of SAM development. I suggest that authors develop a schematic showing the expression patterns/protein accumulation patterns of wus, rev, esr1 and esr2 and draw possible molecular circuitry as suggested by previous studies and the current study.

Author Response

My major concern is that authors do not provide expression pattern of ESR2. Has it been published elsewhere? If so authors must include a schematic and explain how such expression pattern contributed to the rescue of stm mutant phenotype? In general, though genetic analysis is thorough, I struggle to connect this analysis with the possible molecular circuitry of SAM development. I suggest that authors develop a schematic showing the expression patterns/protein accumulation patterns of wus, rev, esr1 and esr2 and draw possible molecular circuitry as suggested by previous studies and the current study.

Thank you for your concern. The expression of ER2 is, indeed, reported previously (Reference 13). In addition, we have included one more article to corroborate the expression pattern of ESR2 (reference 28) in line 254 of the revised manuscript.

As for the expression pattern of STM, WUS, REV, ESR1, and ESR2 protein in the vegetative phase, we have provided the schematic illustration of their expression pattern as Supplementary Figure S3.

At last, an additional Figure is included as Figure 5 to summarize genetic interaction among 5 genes obtained in this study, as well as the previous study.

Reviewer 2 Report

In this manuscript Ikeda et al., investigated the genetic basis underlying lateral organ formation in Arabidopsis. The genes controlling shoot apical meristem and lateral organ formation is well studied in Arabidopsis, but the complex genetic networks linking these genes in these processes are still remained poorly understood. Ikeda et al., studied the genetic interactions between 2 members of AP2/ETHYLENE-RESPONSE FACTOR family (ESR1 and ESR2) and the well-known shoot apical meristem genes. The authors found that ESR1 and ESR2 genetically interact with WUS, STM, REV to control lateral root initiation and adaxial-abaxial polarity. These findings are interesting and might advance our understanding of the lateral organ formation in plants. I have only one comment:

  1. line 103: The authors mentioned that they have identified a new allele of esr mutant (esr1-2) in which the full length of ESR1 is abolished. The authors indicated that esr1-2 is a weaker allele compared to esr1-1 (data not shown). This is a bit surprising because if both alleles are loss-of-function they are expected to exhibit the same phenotype. The authors should check whether the esr1-2 mutant a is a loss-of-function allele? Please check whether the esr2-1 does not retain truncated ESR transcripts that might encode for a functional protein.

Author Response

line 103: The authors mentioned that they have identified a new allele of esr mutant (esr1-2) in which the full length of ESR1 is abolished. The authors indicated that esr1-2 is a weaker allele compared to esr1-1 (data not shown). This is a bit surprising because if both alleles are loss-of-function they are expected to exhibit the same phenotype. The authors should check whether the esr1-2 mutant is a loss-of-function allele? Please check whether the esr2-1 does not retain truncated ESR transcripts that might encode for a functional protein.

Thank you for your concern. In general, not all individual allele exhibits the same phenotypes. For instance, the serrate mutations show a wide range of phenotypes from serrated leaves (weak phenotype observed in se-1 weak allele), defects in adaxial/abaxial leaf polarity in se-2 (intermediate), aberrant shoot and root phenotypes in se-3 (Grigg et al 2005 Supplemental Figure 1_image below). The severity of phenotypes depends on the type of mutation, as well (se-1 mutation causes a frameshift mutation lacking the last 27 amino acids).

To answer your question, we have performed RT-PCR by using primer combination to detect protein coding region of ESR1 transcripts (revised version of Supplementary Figure S1B). We found more abundant accumulation of truncated ESR1 transcripts in esr1-2, however, we are not certain the truncated ESR1 protein accumulation, stability, and the translation efficiency of ESR1 transcripts lacking 3’-UTR. We have described a new result in the corresponding sentence in the manuscript (line 94-96).

As shown in Table 1, esr1-2 mutation moderately enhances phenotypes in the wus-101 mutant background. Similarly, cotyledon phenotypes in the esr1-2 are weaker than those in esr1-1 (manuscript preparation). In addition, we have combined esr1-1 and esr1-2 alleles, termed esr1 transheterozygote, and the resulting esr1 transheterozygote mutant plant exhibit intermediate phenotypes between esr1-2 and esr1-1 (our unpublished result and manuscript preparation), strongly suggesting esr1-2 allele being weak allele.

Round 2

Reviewer 1 Report

The authors have revised the manuscript as suggested.